# The Effect of Disc Surface Topography on the Dry Gross Fretting Wear of an Equal-Hardness Steel Pair

**DOI:** 10.3390/ma12193250

**Published:** 2019-10-04

**Authors:** Agnieszka Lenart, Pawel Pawlus, Andrzej Dzierwa

**Affiliations:** 1EME Aero, Jasionka 954, 36-002 Jasionka, Poland; agnieszka.lenart@eme-aero.com; 2Faculty of Mechanical Engineering and Aeronautics, Rzeszow University of Technology, Powstancow Warszawy 8 Street, 35-959 Rzeszow, Poland; adktmiop@prz.edu.pl

**Keywords:** surface topography, fretting, wear, coefficient of friction

## Abstract

Experimental investigations were carried out with an Optimol SRV5 tribological tester in a flat-on-sphere scheme. The balls co-acted with the discs in a gross sliding fretting regime. The balls and discs were made from the same steel with a very similar hardness. Tests were conducted at 25–35% relative humidity, 30 °C, and a constant normal load and number of cycles (18,000). The discs had different textures after various machining treatments. It was found that the total wear level of the tribological assembly was proportional to the disc surface amplitude. The influence of the disc roughness on the coefficient of friction was evident only for the smallest stroke of 0.1 mm, and the frequency of oscillation affected this dependency.

## 1. Introduction

Fretting is the motion between two contacting parts when the sliding amplitude is small. When the amplitude of oscillation is larger than the radius of elastic contact, reciprocating sliding occurs [1]. Depending on the normal load, the following regimes can be recognized when the sliding amplitude increases: stick, partial slip and gross slip [2,3,4]. Varenberg et al. [5,6] introduced the slip index, which is a criterion to define various fretting regimes. For a low displacement amplitude, a partial slip occurs. When the sliding amplitude is higher, but smaller than a Hertzian contact radius, a gross slip takes place [7]. Fretting regimes are related to the kinds of damage; a material loss occurs in a gross slip, but cracking occurs in a partial slip. Various types of wear occur during fretting: abrasion, adhesion, oxidation and surface fatigue [8].

In fretting, wear oxide debris is formed. Wear particles can have two contrary influences on wear loss; beneficial or harmful [9,10]. Varenberg et al. [8] found that wear debris for dominant adhesion diminished, while for prevailing abrasion, it simplified material loss. Wear debris is also related to the coefficient of friction [11]. Diomidis and Mischler [12] found that an increase in the stroke caused increases in the sizes of wear particles when the fretting stroke increased. During fretting of steel elements, the created wear particles are non-cohesive and loose; therefore, they can be removed without difficulty [13]. 

Many variables (about 50) affect fretting wear [14,15]; a stroke, a normal force, a frequency and surface hardness seem to be the most important.

Fretting wear is proportional to the stroke [16,17]. It was found that when the amplitude of displacement increased, the friction coefficient also increased. Decreasing the normal load reduced the volumetric wear. However, increasing the contact pressure reduced the sliding amplitude, leading to operating in the partial slip regime [18]. Kalin et al. [19] performed dry fretting tests with silicon carbide against bearing steel counterparts. The wear volume increased exponentially with the amplitude. However, in research reported in [12], the coefficient of friction showed no dependence on the amplitude of oscillation, and only the wear rate increased with the stroke.

Several researchers studied the impact of frequency on fretting. When the frequency of oscillation (0.01–60 Hz) was higher, the fretting damage of steel samples was lower [20]. However, Soderberg et al. [21] found that the fretting wear of steel samples was also marginally affected by the frequency (between 10 and 10,000 Hz). Odfalk and Vingsbo [22] found that the amplitude at the transition between gross and partial slips decreased when the frequency was smaller than 300 Hz. Li et al. [23] analysed the influence of the frequency (1–80 Hz) on the contact of steel samples under fretting conditions. They considered changes in accumulated dissipated energy, which is proportional to the wear volume [7,24]. A decrease in this energy and a growth in the maximum coefficient of friction were observed with increasing frequency [23]. Park et al. [25,26] found that an increase in the frequency led to higher wear for contacts between tin-plated and tin-coated brass. The fretting wear of sintered iron was studied by the authors of [27]. Higher frictional resistance corresponded to higher frequency, while the wear level changed in the opposed direction. The fretting wear performance of two magnesium alloys was investigated [28]. The volumetric wear increased with increasing normal load and decreased with increasing frequency; however, the frequency had a marginal effect on the friction coefficient.

The relationship between hardness of co-acting parts and fretting wear was found to be complex. Kayaba and Iwabuchi [29] found that wear of harder steel was higher than that of a softer one, thanks to protection of the harder sample by oxide wear debris. The effect of the hardness of structural steel on wear of the sliding pair—structural steel–bearing steel—was negligible [30]. Budinsky [30] conducted fretting experiments with hard steel against steels of varying hardness. Fretting wear increased when the difference in hardness levels between two counterparts was large. Lemm et al. [15], in contrast to [29,30,31], used the same steel for both counterparts, in which the hardness was varied. For significantly different levels of hardness between two samples, the wear of the harder specimen was larger than that of the softer one. Oxide-based abrasive particles were observed to be embedded into the softer specimen, leading to its protection to wear. A similar effect was observed in tests in which a hard steel contacted a softer aluminum alloy [32,33].

The surface topography of sliding parts probably has a substantial effect on wear particle occurrence in the contact zone because these particles can escape to neighboring valleys when the surface is rough. Therefore increasing the surface roughness height seems to be a method of improving fretting wear resistance [8,34]. However, different effects of the surface texture on fretting have been previously achieved. Kubiak et al. [35,36] achieved a lower friction force for rougher discs. On the contrary, Pawlus et al. [37] found that for a lower roughness height of the disc sample, the total wear of the tribological system and the coefficient of friction were also lower. Yoon et al. [38] studied the effect of ball surface roughness on fretting. Lower friction coefficients in an initial part of the test were observed for the polished (smoother) sphere than for the non-polished (rougher) sphere. The dissipated energy behaved similar to the maximum coefficient of friction. Li et al. [23] found that the accumulated dissipated energy for a sliding pair with a laser-polished sphere was larger than that for the non-polished sphere. Raeymaekers and Talke [18] observed that the total dissipated energy was marginally higher for the laser-polished sphere than for the regular hemisphere. The authors of [39] found that differences in the surface texture affected the fretting wear between two Zn–Ni coatings. Rougher coatings had much deeper wear scars and more oxidized wear debris than smoother coatings. In [40], the fretting wear of Al–Si alloys after various surface preparations was studied. An increase in the surface roughness height led to improved wear resistance. The authors of [41] investigated the influence of the surface texture of steel discs contacting ceramic balls on the tribological performance of the sliding assembly in a dry gross sliding fretting regime and found that the results were better for lower disc roughness heights. The influence of disc surface texturing on dry gross sliding fretting of a ball-on-disc assembly was investigated [42]. The effect of surface texturing depended on the dominant wear type (adhesion or abrasion). Surface roughness affected the friction coefficient at the transition between partial and gross slips; for lower heights, this coefficient was higher [43,44]. Leonard et al. [45] modelled fretting wear using the combined finite-discrete element method. A rougher surface led to higher wear. However, the role of the initial surface roughness in the gross slip regime seems to be important only at the beginning of tests.

One can see from this review that the effect of the surface topography of contacting elements on the tribological performances of sliding pairs in gross sliding fretting regimes is still unclear. 

## 2. Materials and Methods 

Tests were conducted using an Optimol SRV5 tribological tester (Optimol Instruments Prüftechnik GmbH, Munich, Germany) in a flat-on-ball configuration. The previous versions of this tester were used in other fretting experiments [16,18,36]. Balls of 5 mm radius R slid against discs in a dry gross sliding fretting regime (fulfilling the criteria contained in [1,4,5]). Balls and discs were made from 100Cr6 steel, of equal hardness (60 HRC). Experiments were conducted at a temperature of 30 °C, at 25–35% relative humidity. The number of cycles (18,000) was constant. Table 1 presents the experimental conditions. Before each test, samples were cleaned in acetone. The minimum number of test repetitions was three.

Surface topographies of machined and worn counterparts were measured by a Talysurf CCI Lite scanning coherent interferometer (Taylor Hobson, Leicester, UK). The measured areas were 3.3 × 3.3 mm^2^, and sampling intervals in perpendicular directions were 3.2 μm. The average value of the Ra parameter of machined balls was 0.15 µm. Discs were prepared by various methods including polishing (P), lapping (L), milling (M), grinding (G) and vapour blasting (VB). Tribological tests of anisotropic disc topographies (M and G) were conducted perpendicularly to lays (main directions). The wear volume V_tot_ of the whole tribological system was computed as Equations (1)–(3):V_tot_ = V_disc_ + V_ball_,(1)
V_disc_ = (V_disc−_) − (V_disc+_),(2)
V_ball_ = (V_ball−_) − (V_ball+_),(3)
where volumes (V_disc+_) and (V_ball+_) were buildups or materials transferred; volumes (V_disc−_) and (V_ball−_) were lost materials [13,32].

The wear levels of discs were assessed only after surface leveling, and a digital filtration was not used. Before the calculation of wear levels of the balls, their curvatures were eliminated using spheres.

Table 2 shows selected surface topography parameters of discs before wear: the root mean square height Sq, the texture aspect ratio Str, the autocorrelation length Sal, the skewness Ssk, the kurtosis Sku and the root mean square slope Sdq [46]. Figure 1 shows contour maps of the disc surfaces.

The heights (Sq) and slopes (Sdq) of isotropic (the Str parameter in the range: 0.85–0.87) vapour-blasted surfaces VB1 and VB2 were the biggest. The surface height of the sample VB2 after finish vapour blasting was smaller. Anisotropic samples after finish milling M1 and M2 had radial structures, but after grinding G1 and G2, they had one-directional structures. Amongst the anisotropic surfaces, the M1 sample was the roughest and had the largest autocorrelation length Sal. The height of the finish ground surface G2 was smaller than that of the rough ground G1 texture. The heights of the milled M2 and ground G1 samples G1 were similar. The correlation lengths of samples after finish milling were larger than those of discs after grinding. The roughness heights of polished P and lapped L samples were the lowest. The surfaces after milling had deterministic characters, while other surfaces had random characters.

The surfaces of worn counterparts were examined by scanning electron microscope (SEM) fitted out with an energy dispersive spectrometry (EDS) analyser. 

## 3. Results and Discussion

Table 3 lists the results of the fretting tests. It shows the volumetric wear levels of discs and balls, and the mean and final coefficients of friction. There were 18,000 cycles, corresponding to a test duration of 15 min for the frequency of 20 Hz, and 6 min for the frequency of 50 Hz. When the frequency was 20 Hz, the average value of the largest friction coefficient (COF_50-900_) was estimated after removal of the early parts of the results (rapid growths of the coefficient of friction) before 50 s of the test, while the final value (COF_600-900_) was assessed as the mean from the last five minutes of the test. When the frequency was set to 50 Hz, COF_50-360_ characterized the mean and COF_300-360_ characterized the final friction coefficient.

The maximum surface elastic contact pressure *p_o_* was calculated by Equation (4) [47]:(4)p0=(6PE*2π3R2)13.

The radius of elastic contact length *a* was computed by the following Equations (5) and (6) [47]:(5)a=(3PR4E*)13.
where:(6)$$E^{*}={\left(\frac{1-\nu_{1}{}^{2}}{E_{1}}+\frac{1-\nu_{2}{}^{2}}{E_{2}}\right)}^{-1}$$

*E*_1,2_ = Young’s moduli, *ν*_1,2_ = Poisson’s ratios.

When the normal force was 45 N, the Hertzian radius of contact *a* was 0.114 mm; it was higher than the largest amplitude of oscillation (0.1 mm), and therefore fretting occurred. In this case, the maximum elastic contact pressure *p_0_* was 1652 MPa.

The variation of the friction coefficient was typically not higher than 0.03. Scatters of total volumetric wear were usually not higher than 14% for assemblies with the same disc samples. Table 3 lists the results of experimental investigations. 

In test A, the wear of the discs was typically higher than the wear of the balls, but the sliding pair with the roughest disc sample VB1 was the exception. The final friction coefficient COF_600-900_ was similar for various assemblies; however, the average friction coefficient COF_50-900_ was the smallest for the sliding pair containing the polished disc P of the lowest roughness height. The friction coefficient initially increased and then obtained a stable value typically after 15,000 cycles (Figure 2a). The total volumetric wear level was proportional to the disc surface height determined by the Sq parameter; the coefficient of linear correlation was 0.95 (Figure 3a).

In test B, for the sliding pair containing disc VB1, the wear rate of the disc was lower than that of the ball. For different sliding pairs, the opposed situation was observed. The wear of the balls was inversely proportional to the wear of the discs. Similarly to test A, the friction coefficient after initial sharp growth, increased slowly and obtained a stable value in most cases after 15,000 cycles (Figure 2b). The smallest friction coefficient was achieved for the assembly containing the disc after lapping L, while the highest coefficient was acheived for sliding pairs with discs after polishing P, milling M1, and grinding G2. A decrease in the amplitude of oscillation from 0.1 (test A) to 0.075 mm caused a reduction in volumetric wear, but had a marginal effect on the coefficient of friction. Total wear was proportional to the standard deviation of the disc height; the linear coefficient of correlation was 0.85 (Figure 3b).

Figure 4 shows the runs of the maximum coefficients of friction during tests C for sliding pairs containing all selected discs. The shapes of the presented curves are similar to those obtained in tests A and B. In all analysed cases, the wear levels of the balls were much lower than the wear levels of the discs. Similar to the assemblies analysed, the previous wear of the system was proportional to the disc roughness height (Figure 5a). The average coefficients of friction were higher for bigger disc topography heights (Figure 5b). The roughest disc sample VB1 led to the highest total wear level and the average friction coefficient COF_50-900_. The final values of the coefficient of friction COF_600-900_ were similar for various sliding pairs (between 0.97 and 1.01). A reduction in the stroke from 0.15 (test B) to 0.1 mm led to smaller volumetric wear without a change in the final coefficient of friction.

In test D, contrary to the results analysed previously, the average wear rates of the balls were similar to those of the discs. The assembly with the roughest disc VB1 led to the biggest, while the smoothest disc P, led to the smallest wear volume of the tribological system. The wear of the balls and the total wear were proportional to the disc roughness height. The highest coefficient of friction was obtained for the roughest sample VB1, while the smallest was obtained for sample M2 after milling. The friction forces increased during tests and obtained steady values typically after 9000 cycles (Figure 6a). A reduction in the contact pressure for the same frequency of oscillations (test C) caused a slight wear decrease and growth in the friction coefficient.

In test E, the wear of the balls was typically lower than the wear of the discs, but assemblies with VB1 and G1 discs were the exceptions. Similar to other tests, the volumetric wear was proportional to the disc roughness height. The total wear of the sliding pair with the roughest disc sample VB1 was the biggest, but with the smoothest disc P, it was the lowest. The wear of the balls was inversely related to the wear of the discs. The highest mean friction coefficient was obtained for the sliding pair with the polished sample of the smallest height P, while the smallest was obtained for the assemblies with ground samples G1 and G2. It is interesting that the M2 sample after milling led to comparatively high frictional resistance, contrary to test D carried out at the same load and smaller frequency of oscillation. Similar to test D, the friction coefficient after early fluctuations typically obtained stable values after 9000 cycles (Figure 6b). A growth in the frequency of oscillations for the same load of 15 N (test D) caused wear reduction and growth in the friction coefficient.

In test F, except for assembly with the vapour-blasted VB1 sample, the wear rates of the discs were larger than the wear rates of the balls. The wear of the balls was inversely proportional to the wear of the discs. In most cases, the negative wear levels of the balls were obtained, which means that build-ups or material transfers were formed. The total wear level of the assembly with the roughest VB1 sample after vapour blasting was the largest, but with the smoothest P sample after polishing, it was the smallest. As a result, the total wear and wear rates of the balls were proportional to the Sq parameter of the disc textures. It is also interesting that the roughest disc sample VB1 yielded the smallest coefficient of friction from all analysed assemblies. In this case, the scatter of the average friction coefficient did not coincide with scatters corresponding to the other sliding pairs, except for the assembly containing the milled M1 specimen. Disc sample M2 after milling led to the highest mean friction coefficient. Figure 7 presents values of the maximum friction coefficient runs for sliding pairs with all and chosen discs in test F. The coefficient of friction after initial abrupt growth was stable between 2000 and 7000 cycles and then increased as the test progressed. The results presented in Figure 7b (the lowest friction coefficient of the assembly with VB1 disc) are different from those shown in Figure 4b obtained for the same normal load and smaller frequency of oscillations (test C). However, the change in the frequency for the same normal load caused a marginal change in total wear volume. An increase in the normal load for the same frequency of oscillations (test E) led to wear growth and a reduction in the coefficient of friction.

Figure 8 shows fretting loops for the sliding pair with a lapped L disc. One can observe changes in the relative displacements without changes in the coefficients of friction during the analysis of the fretting loops corresponding to tests A, B, C. After comparing fretting loops obtained after tests C and D, as well as F and E, it is evident that a reduction in the load led to growth in the friction coefficient and a change in the loop shape corresponding to a higher value of the slip index. An increase in the frequency from 20 (tests A, B, C and D) to 50 Hz (tests E and F) led to changes in fretting loop shapes (smaller stability of the coefficient of friction at a higher frequency). 

Figure 9 and Figure 10 show contour maps of vapour-blasted VB1 and polished P discs and contacted balls, respectively, after various tests. It is evident that smaller volumetric wear occurred for the sliding pair containing the P disc compared to the VB1 disc. For assembly with VB1, the disc wear of the disc was typically lower than the wear of the ball, contrary to the assembly with the P disc. Wear of co-acting pairs had an abrasive-adhesive character. Plastic deformation also occurred on the vapour-blasted VB1 disc surface.

The influence of the disc surface texture on the frictional resistance was evident mainly at the lowest amplitude of motion (0.05 mm), when wear of the disc was comparatively low. The effect of the disc surface texture on the friction coefficient depended on the frequency of oscillation. For the normal force of 45 N and the frequency of 20 Hz (test C), the highest disc roughness (VB1 sample) led to the largest mean coefficient of friction; however, the opposed tendency occurred for the same load and the frequency of 50 Hz (test F). For the lower normal force of 15 N, a similar effect of frequency on friction force was not as apparent; however, in test E (frequency of 50 Hz), the highest mean friction coefficient was obtained for the assembly containing the disc P with the smallest height. 

The total wear rate was proportional to the height of the disc surface, determined by the value of the Sq parameter, independently of the operating conditions. In tests A, C, D, E and F, the total wear of assembly with the roughest vapour-blasted VB1 sample was the highest, it was the smallest with the smoothest polished specimen P. This performance probably resulted from the fact that rougher surfaces, of higher plasticity indices [48,49], have a larger tendency to plastic deformation and hence wear. Tracks of the plastic deformation were found on the VB1 disc surface after tribological tests (Figure 9). The wear level of balls was typically smaller than of discs, but the assembly with the roughest VB1 disc was the exception.

Previously, experiments were conducted using the same tester [50]. Steel balls made of 100Cr6 steel of 60 HRC hardness contacted discs of various surface topographies made of 42CrMo4 steel of 40 HRC hardness under the same conditions as in the tests A, B, C. In the previous tests the resistance to motion was the smallest when the movement of the ball was orthogonal to the main direction of the finished milled disc surface. These results were probably related to formation of the layer of oxidized wear particles on the disc surface and the smaller wear of discs compared to the wear of balls. However, in the present tests, this effect disappeared. Oxidised abrasive particles were embedded into both counterparts (Figure 11 and Figure 12). A low coefficient of friction of assembly with the milled disc surface M2 was achieved only in test D. An increase in hardness of the disc compared to the results presented in [50] led to a smaller role of the oxidized debris layer, and hence, smaller wear of the tribological system and typically smaller wear of the balls compared to that of discs. A decrease in total wear of a sliding pair due to a decrease in hardness differences was found in the other works [14,31].

A decrease in the stroke led to wear reduction. A reduction in the contact pressure led to growth in the friction coefficient and a reduction in the wear volume. An increase in the frequency of oscillation led to an increase in the coefficient of friction and a reduction in volumetric wear only when a lower normal force was applied (15 N). Similar results were obtained for different hardness of counterparts [51].

## 4. Conclusions

For equal-hardness pairs, the wear of the disc was typically higher than that of the ball. Total wear of the tribological system was proportional to the disc roughness height. The influence of the disc surface texture on the friction coefficient was substantial only for the smallest stroke of 0.1 mm. The frequency of oscillation had an effect on this dependence. For the normal load of 45 N and the frequency of 20 Hz, the highest disc roughness corresponded to the largest mean coefficient of friction; however, after increasing the frequency to 50 Hz, the biggest disc amplitude led to the smallest frictional resistance. 

A reduction in the amplitude of oscillation decreased wear but had marginal influence on the coefficient of friction. The friction coefficient increased and total wear decreased due to a reduction in the normal load. When the frequency increased, wear decreased and the friction coefficient increased. These effects were visible only when a lower normal load was applied.

## Figures and Tables

**Figure 1 materials-12-03250-f001:**
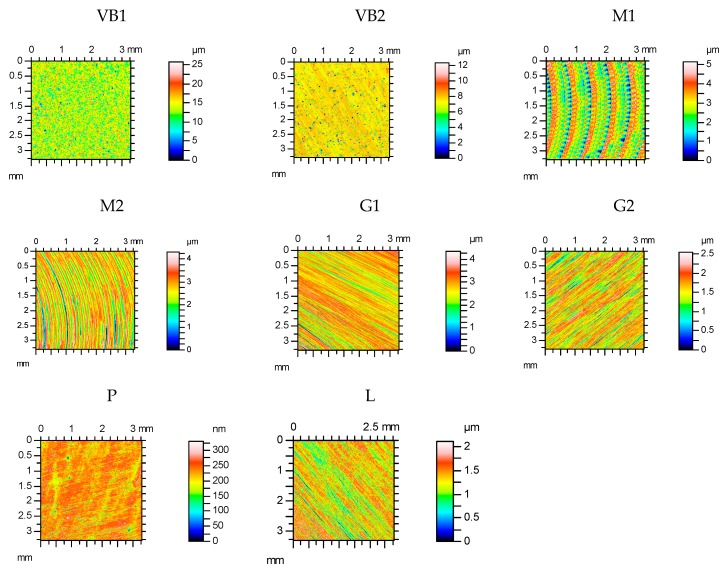
Contour maps of disc surfaces: after vapour blasting VB1 and VB2, after milling M1 and M2, after grinding G1 and G2, after polising P and after lapping L.

**Figure 2 materials-12-03250-f002:**
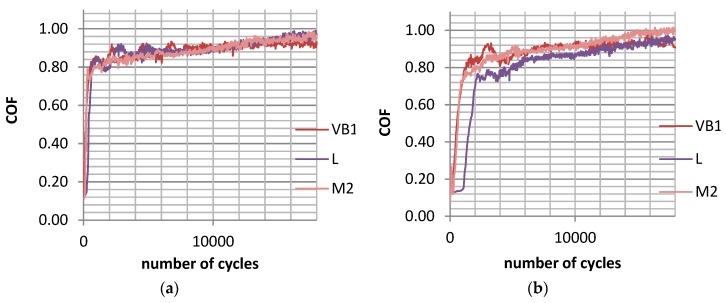
Maximum coefficient of friction during test A; s = 0.2 mm, P = 45 N, f = 20 Hz (**a**) and during test B; s = 0.15 mm, P = 45 N, f = 20 Hz (**b**) for selected assemblies with discs: after vapour blasting VB1, after lapping L and after milling M2.

**Figure 3 materials-12-03250-f003:**
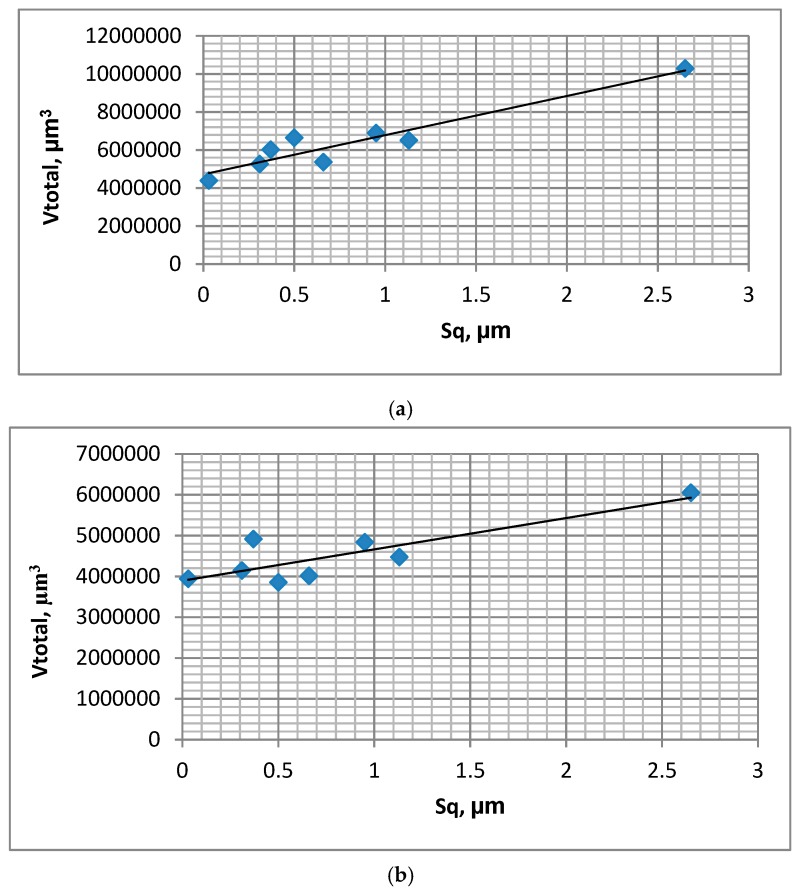
Dependence between disc surface height determined by the Sq parameter and total volumetric wear of tribological system V_total_ during tests A; s = 0.2 mm, P = 45 N, f = 20 Hz (**a**) and B; s = 0.15 mm, P = 45 N, f = 20 Hz (**b**).

**Figure 4 materials-12-03250-f004:**
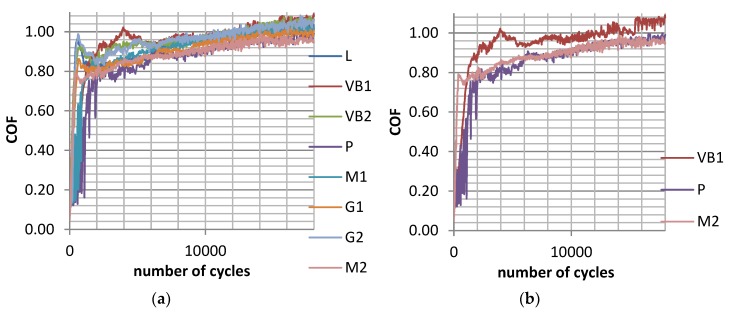
Maximum coefficient of friction during test C for assemblies with all (**a**) and selected discs after vapour blasting VB1, after lapping L and after milling M2 (**b**) for the following operating parameters: s = 0.1 mm, P = 45 N, f = 20 Hz.

**Figure 5 materials-12-03250-f005:**
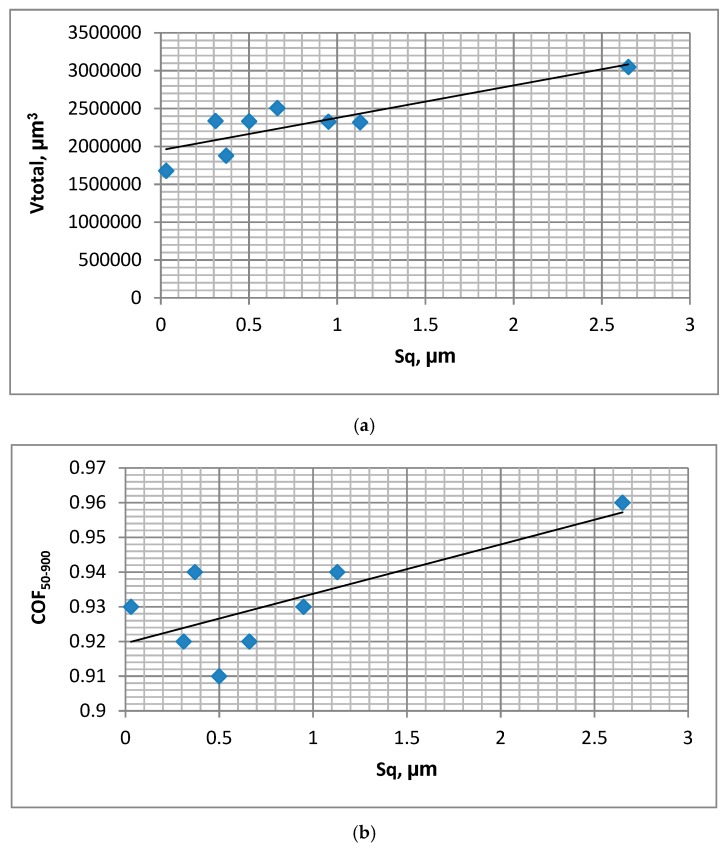
Dependencies between disc surface height determined by the Sq parameter and total volumetric wear of system V_total_ (**a**) and the mean coefficient of friction (**b**) in test C: s = 0.1 mm, P = 45 N, f = 20 Hz.

**Figure 6 materials-12-03250-f006:**
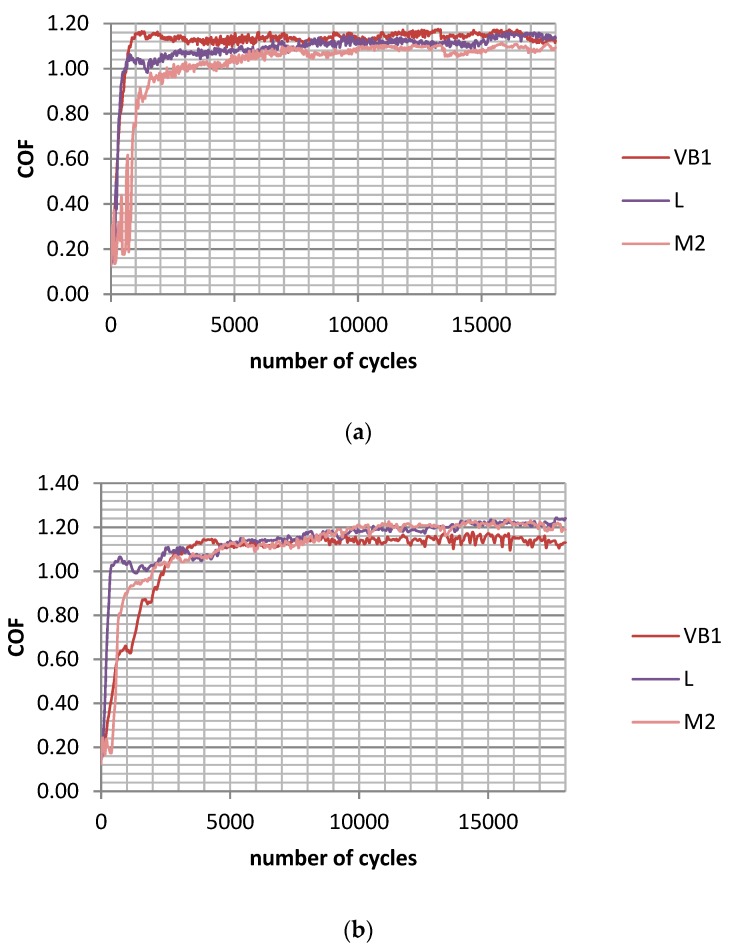
Maximum coefficient of friction during test D; s = 0.1 mm, P = 15 N, f = 20 Hz; (**a**) and during test E; s = 0.1 mm, P = 15 N, f = 50 Hz (**b**) for selected assemblies with discs: after vapour blasting VB1, after lapping L and after milling M2.

**Figure 7 materials-12-03250-f007:**
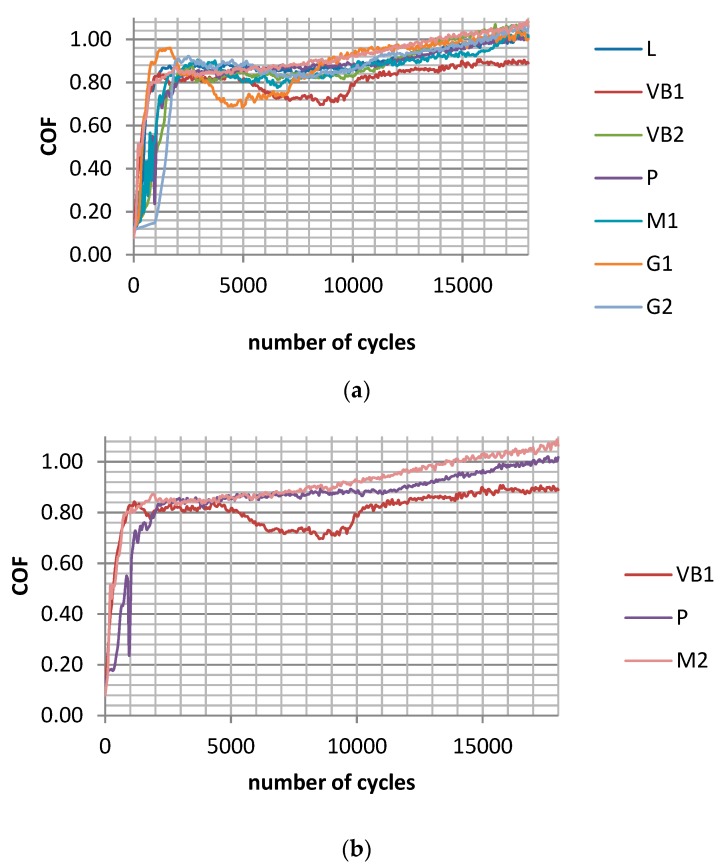
Maximum coefficient of friction during test F for sliding pairs with all (**a**) and selected discs after vapour blasting VB1, after lapping L and after milling M2 (**b**) for the following operating parameters: s = 0.1 mm, P = 45 N, f = 50 Hz.

**Figure 8 materials-12-03250-f008:**
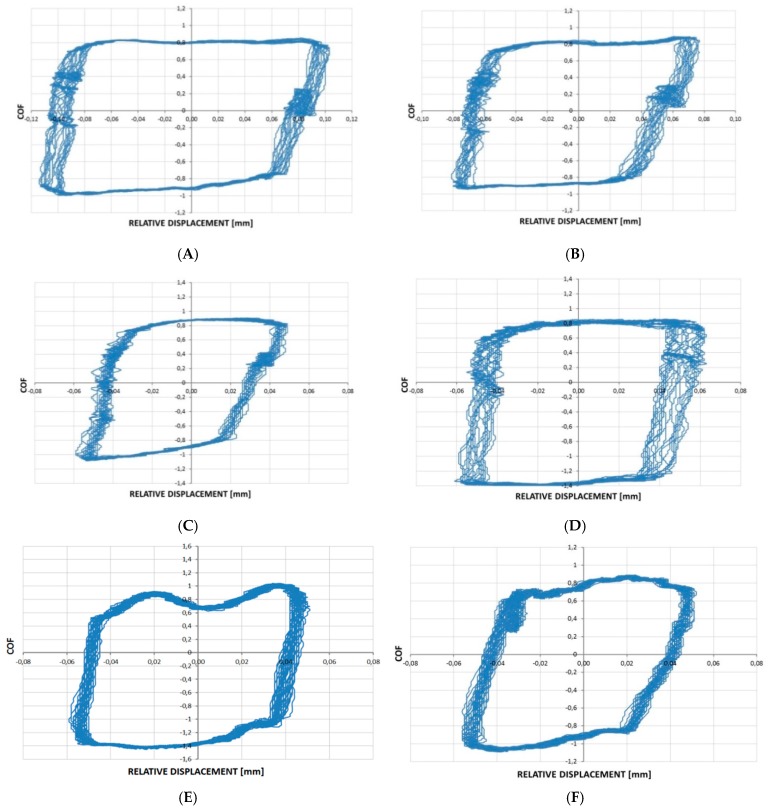
Fretting loops for sliding pair with lapped disc L after tests: (**A**) s = 0.2 mm, P = 45 N, f = 20 Hz, s = 0.2 mm, P = 45 N, f = 20 Hz; (**B**) s = 0.15 mm, P = 45 N, f = 20 Hz; (**C**) s = 0.1 mm, P = 45 N, f = 20 Hz; (**D**) s = 0.1 mm, P = 15 N, f = 20 Hz; (**E**) s = 0.1 mm, P = 15 N, f = 50 Hz and (**F**); s = 0.1 mm, P = 45 N, f = 50 Hz.

**Figure 9 materials-12-03250-f009:**
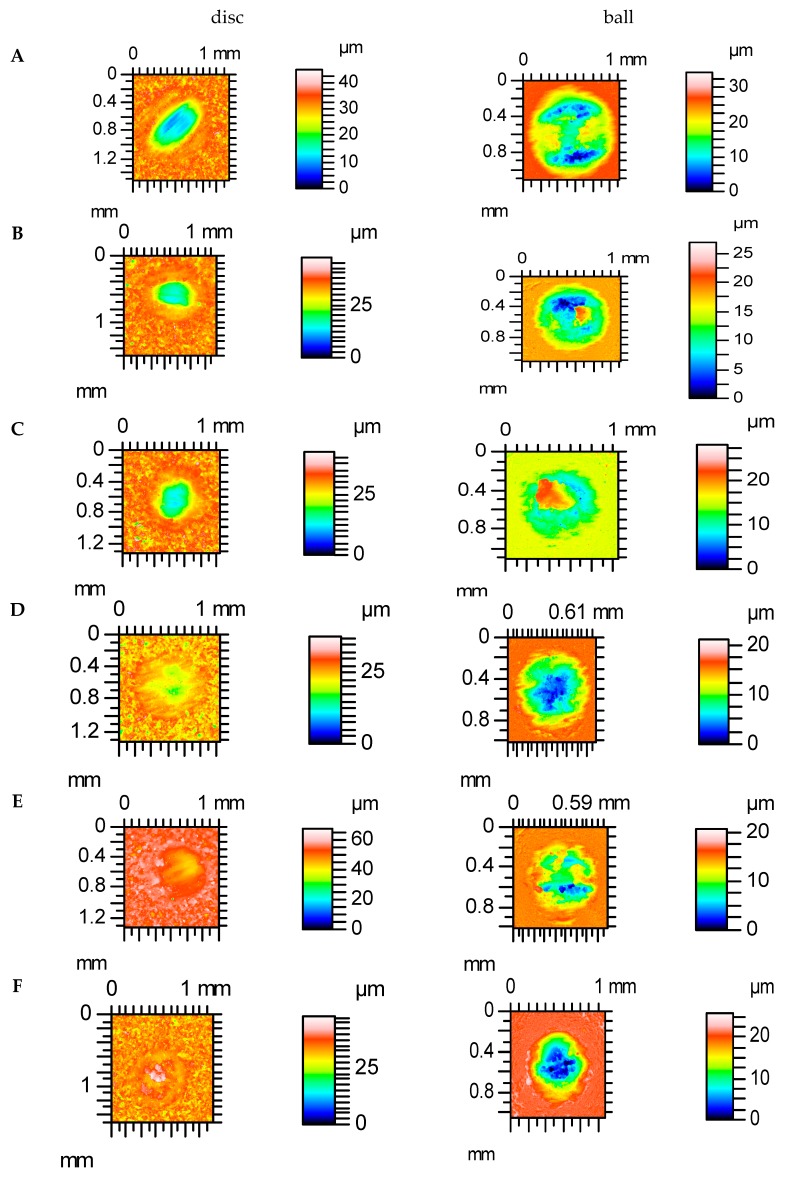
Contour maps of disc vapour-blasted VB1 and contacted balls after tests: (**A**–**F**).

**Figure 10 materials-12-03250-f010:**
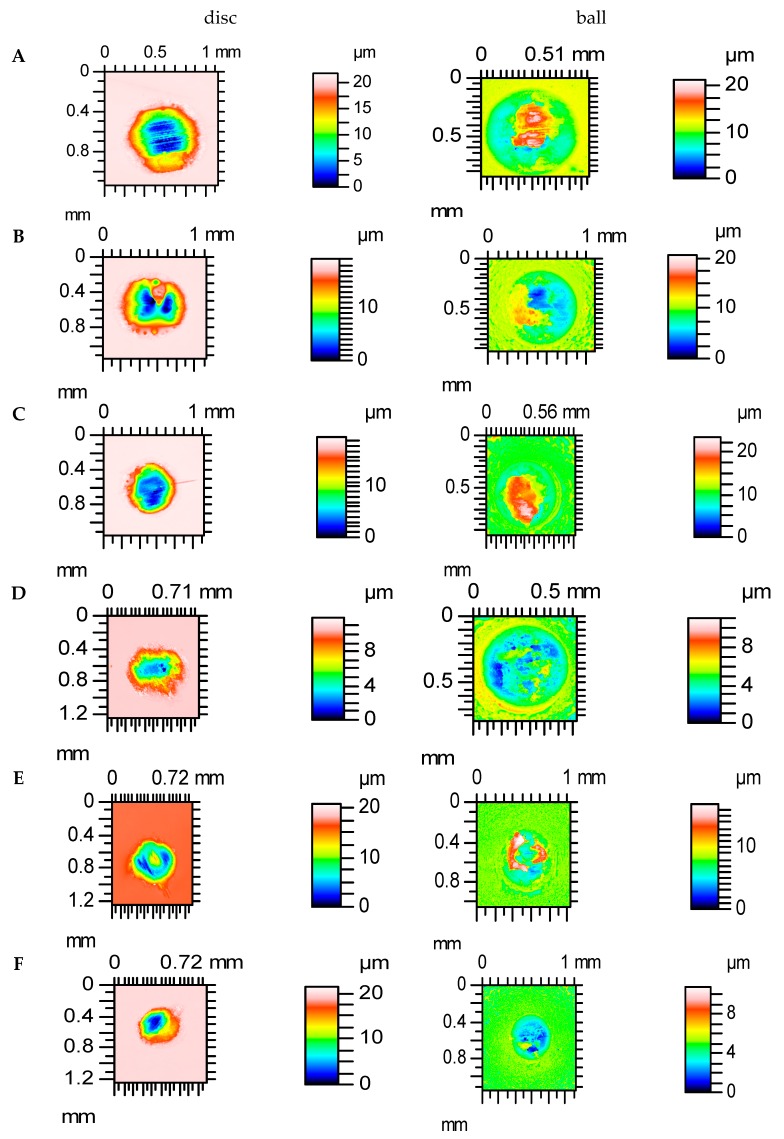
Contour maps of polished disc P and contacted balls after tests: (**A**–**F**).

**Figure 11 materials-12-03250-f011:**
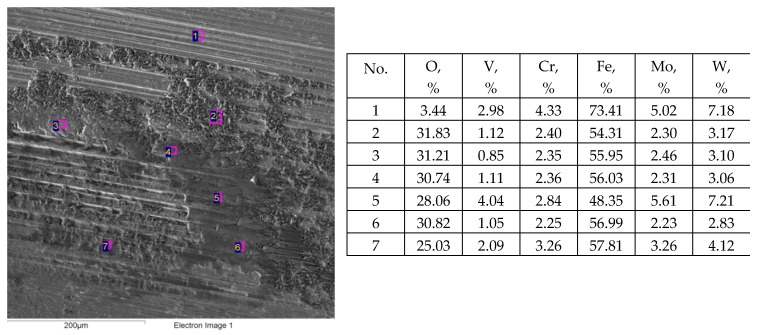
Energy dispersive spectrometry (EDS) analysis of worn lapped disc L after test A; s = 0.2 mm, P = 45 N, f = 20 Hz.

**Figure 12 materials-12-03250-f012:**
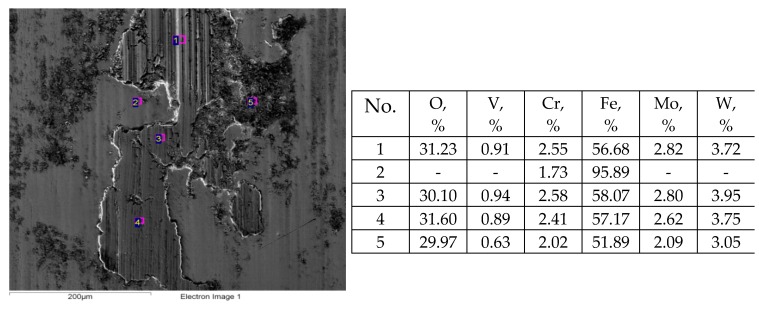
EDS analysis of the ball co-acted with worn lapped disc L after test A; s = 0.2 mm, P = 45 N, f = 20 Hz.

**Table 1 materials-12-03250-t001:** Experimental conditions of gross sliding fretting tests.

Test Designation	Stroke s, mm	Frequency f, Hz	Normal load P, N
A	0.2	20	45
B	0.15	20	45
C	0.1	20	45
D	0.1	20	15
E	0.1	50	15
F	0.1	50	45

**Table 2 materials-12-03250-t002:** Parameters of surface topographies of discs: Sq, Ssk, Sku, Sal, Str and Sdq before fretting tests.

Surface	Parameters					
	Sq, µm	Ssk	Sku	Sal, mm	Str	Sdq
VB1	2.65	−0.5	4.6	0.026	0.872	0.478
VB2	1.13	−0.9	6.6	0.021	0.851	0.193
M1	0.95	−0.25	2.7	0.07	0.205	0.085
M2	0.66	−0.5	3.86	0.034	0.118	0.074
G1	0.5	−0.7	5.5	0.019	0.036	0.118
G2	0.37	−0.56	3.96	0.023	0.046	0.076
P	0.03	−0.8	7.1	0.0465	0.448	0.0082
L	0.31	−0.57	2.9	0.02	0.0154	0.0742

**Table 3 materials-12-03250-t003:** Results of tribological tests; V_disc_—volumetric wear of disc, V_ball_—volumetric wear of ball, COF—friction coefficient.

Test Designation	Disc Surface	V_ball−_, µm^3^	V_ball+_, µm^3^	V_disc−_, µm^3^	V_disc+_, µm^3^	V_ball_, µm^3^	V_disc_, µm^3^	V_total_, µm^3^	COF_50-900_	COF_600-900_
A s = 0.2 mm f = 20 Hz P = 45 N	VB1	5,751,187	192,796	4,735,133	7436	5,558,391	4,727,697	10,286,088	0.89	0.92
	VB2	1,761,967	344,568	5,099,394	8894	1,417,399	5,090,500	6,507,899	0.9	0.95
	M1	2,543,358	144,203	4,518,717	17,577	2,399,155	4,501,140	6,900,295	0.89	0.94
	M2	958,319	937,410	5,350,101	5519	20,909	5,344,582	5,365,491	0.89	0.94
	G1	2,299,195	211,401	4,588,955	31,448	2,087,794	4,557,507	6,645,301	0.88	0.93
	G2	1,686,781	327,076	4,675,078	17,311	1,359,705	4,657,767	6,017,472	0.89	0.93
	P	1,089,701	364,045	3,669,498	973	725,656	3,668,525	4,394,181	0.85	0.92
	L	1,219,342	403,471	4,459,393	8894	815,871	4,450,499	5,266,370	0.9	0.95
B s = 0.15 mm f = 20 Hz P = 45 N	VB1	3,258,026	93,397	2,890,038	7436	3,164,629	2,882,602	6,047,231	0.9	0.93
	VB2	1,484,837	1,049,037	4,043,784	8137	435,800	4,035,647	4,471,447	0.88	0.95
	M1	1,140,223	80,164	3,808,204	29,495	1,060,059	3,778,709	4,838,768	0.96	1.02
	M2	750,209	1,151,923	4,421,414	4302	−401,714	4,417,112	4,015,398	0.91	0.98
	G1	846,395	550,909	3,590,759	32,553	295,486	3,558,206	3,853,692	0.89	0.95
	G2	1,181,852	18,356	3,760,894	11,346	1,163,496	3,749,548	4,913,044	0.96	1
	P	1,333,566	119,097	2,725,342	14	1,214,469	2,725,328	3,939,797	0.96	1.02
	L	854,582	359,220	3,660,324	10,598	495,362	3,649,726	4,145,088	0.86	0.93
C s = 0.1 mm f = 20 Hz P = 45 N	VB1	1,245,437	384,503	2,217,610	28,833	860,934	2,188,777	3,049,711	0.96	1
	VB2	1,050,962	67,675	1,373,249	37,254	983,287	1,335,995	2,319,282	0.94	1
	M1	707,693	65,643	1,701,742	16,530	642,050	1,685,212	2,327,262	0.93	1.01
	M2	519,194	442,163	2,438,983	9360	77,031	2,429,623	2,506,654	0.92	0.98
	G1	518,063	367,950	2,218,546	37,754	150,113	2,180,792	2,330,905	0.91	0.97
	G2	340,822	419,387	1,969,885	12,898	−78,565	1,956,987	1,878,422	0.94	1
	P	201,539	916,405	2,393,498	227	−714,866	2,393,271	1,678,405	0.93	1
	L	465,848	239,594	2,129,954	19,625	226,254	2,110,329	2,336,583	0.92	0.98
D s = 0.1 mm f = 20 Hz P = 15 N	VB1	2,854,881	11,649	785,083	431,423	2,843,232	353,660	3,196,892	1.12	1.13
	VB2	1,179,948	21,418	1,047,159	117,925	1,158,530	929,234	2,087,764	1.08	1.12
	M1	1,508,291	26,515	1,317,243	20,301	1,481,776	1,296,942	2,778,718	1.1	1.13
	M2	1,011,996	12,048	881,287	17,653	999,948	863,634	1,863,582	1.06	1.09
	G1	1,326,692	12,452	814,496	46,223	1,314,240	768,273	2,082,513	1.1	1.115
	G2	654,536	20,406	1,034,300	14,157	634,130	1,020,143	1,654,273	1.09	1.12
	P	426,287	25,047	987,545	1284	401,240	986,261	1,387,501	1.135	1.15
	L	906,603	44,715	1,119,316	812	861,888	1,118,504	1,980,392	1.09	1.12
-	-	-	-	-	-	-	-	-	COF_50-360_	COF_300-360_
E s = 0.1 mm f = 50 Hz P = 15 N	VB1	3,227,445	33,450	285,173	865,175	3,193,995	−580,002	2,613,993	1.12	1.13
	VB2	805,101	7064	917,141	2819	798,037	914,322	1,712,359	1.11	1.16
	M1	642,892	19,212	941,479	1882	623,680	939,597	1,563,277	1.14	1.16
	M2	176,842	616,892	1,112,931	1816	−440,050	1,111,115	671,065	1.14	1.2
	G1	1,687,468	28,080	701,315	75,070	1,659,388	626,245	2,285,633	1.08	1.12
	G2	472,249	50,237	635,830	90,243	422,012	545,587	967,599	1.09	1.12
	P	291,999	12,621	380,238	35,335	279,378	344,903	624,281	1.18	1.21
	L	273,365	131,520	953,558	11,421	141,845	942,137	1,083,982	1.16	1.21
F s = 0.1 mm f = 50 Hz P = 45 N	VB1	3,221,366	17,892	1,657,568	576,659	3,203,474	1,080,909	4,284,383	0.85	0.94
	VB2	368,583	632,212	2,487,315	67,377	−263,629	2,419,938	2,156,309	0.92	1.07
	M1	421,753	638,573	2,470,523	44,431	−216,820	2,426,092	2,209,272	0.89	1.02
	M2	148,466	964,172	2,830,504	7317	−815,706	2,823,187	2,007,481	0.96	1.07
	G1	591,749	604,496	2,184,448	19,451	−12,747	2,164,997	2,152,250	0.9	1.03
	G2	187,920	835,974	2,420,636	3788	−648,054	2,416,848	1,768,794	0.91	1.05
	P	86,437	875,835	2,376,765	578	−789,398	2,376,187	1,586,789	0.93	1
	L	475,883	415,024	2,366,206	650	60,859	2,365,556	2,426,415	0.895	1

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
