# Peer review of "The Effect of Disc Surface Topography on the Dry Gross Fretting Wear of an Equal-Hardness Steel Pair"

_materials, 2019, doi:10.3390/ma12193250_

Round 1

Reviewer 1 Report

I do not have any comments concerning professional content of this article. 

I recommend publishing of the article in present form.

Author Response

Dear Reviewer,

Thank you for your opinion.

Reviewer 2 Report

Dear authors, please see my comments in the attached file.

Author Response

Dear Reviewer,

Thank you for your comments.

The title of the MS does not reflect its subject; in my opinion the term “fretting wear” is more suitable than “fretting” alone, in any case the title of the MS needs to be rewritten in order to reflect adequately the subject of the work.

The title has been changed.

In lines 29-30 of the MS the reference [9] is missing.

Improved.

I think that ref. [14] should be the following:

Dobromirski J. Variables of Fretting Process: Are There 50 of Them? Stand. Frett. Fatigue Test Methods Equip., 100 Barr Harbor Drive, PO Box C700, West Conshohocken, PA 19428-2959: ASTM International; 2009, p. 60-60–7. doi:10.1520/STP25816S.

This reference has been added.

In the MS the term “gross fretting conditions” is continuously used, I think it is more appropriate to use “gross sliding fretting conditions”.

Changed accordingly.

The expressions, and the corresponding reference, for the theoretical formulae of the maximum surface normal pressure, p0, and radius of contact length, a, must be included in the MS.

These expressions have been added.

In Table 3 the operational parameter (amplitude, frequency and normal force) corresponding for each test type must be included; these data ease the analysis of the results presented in this table.

Done.

The contours plots shown in Figures 9 and 10 are too small, please optimise the blanc space in the Figure in order to make these plots more visible.

These contour plots have been improved.

Reviewer 3 Report

Authors investigated the effect of disc surface topography on dry gross fretting of equal-hardness steel pair. It is interesting to point out that total wear of the tribological system was proportional to a disc roughness height. This manuscript is suitable to publish in Materials. There are some concerns to be addressed:

Please describe more detail for each caption of Table and Figure. For Fig. 4a and Fig. 7a, it’s hard to distinguish each curve. Please replot them. Please provide SEM and EDX data for disc and co-acted balls. Please indicate the influence of surface roughness on the wear rate. Please discuss tribological tests in depth. The manuscript should be written in professional English.

Author Response

Dear Reviewer,

Thank you for your comments.

Please describe more detail for each caption of Table and Figure. For Fig. 4a and Fig. 7a, it’s hard to distinguish each curve. Please replot them.

All figures presented the dependence between the number of cycles and the coefficient of friction have been improved. Captions of all figures have been extended.

Please provide SEM and EDX data for disc and co-acted balls.

Done.

Please indicate the influence of surface roughness on the wear rate.

The discussion about the effect of surface roughness on the wear rate was extended.

Please discuss tribological tests in depth.

Done.

The manuscript should be written in professional English.

English has been improved.

Round 2

Reviewer 3 Report

Accept